# Comparison of Adverse Effects of Two SARS-CoV-2 Vaccines Administered in Workers of the University of Padova

**DOI:** 10.3390/vaccines11050951

**Published:** 2023-05-05

**Authors:** Paola Mason, Rosario Rizzuto, Luca Iannelli, Flavio Baccaglini, Valerio Rizzolo, Andrea Baraldo, Barbara Melloni, Francesca Maffione, Camilla Pezzoli, Maria Laura Chiozza, Giampietro Rupolo, Marco Biasioli, Filippo Liviero, Maria Luisa Scapellato, Andrea Trevisan, Stefano Merigliano, Alberto Scuttari, Angelo Moretto, Bruno Scarpa

**Affiliations:** 1Department of Cardiac, Thoracic, Vascular Sciences and Public Health, University of Padova, Via Giustiniani 2, 35128 Padova, Italy; 2Department of Biomedical Sciences, University of Padova, Via Ugo Bassi 58/B, 35131 Padova, Italy; 3Central Administration, University of Padova, Via VIII Febbraio 2, 35122 Padova, Italy; 4Italian Red Cross, Padua Committee, Via della Croce Rossa, 130, 35129 Padova, Italy; 5Department of Surgery, Oncology and Gastroenterology, University of Padova, Via Giustiniani 2, 35128 Padova, Italy; 6Department of Statistical Sciences, University of Padova, Via Battisti 241, 35121 Padova, Italy

**Keywords:** side-effects, COVID-19 vaccines, workers, education sector

## Abstract

**Introduction**: In Italy, on December 2020, workers in the education sector were identified as a priority population to be vaccinated against COVID-19. The first authorised vaccines were the Pfizer-BioNTech mRNA (BNT162b2) and the Oxford-AstraZeneca adenovirus vectored (ChAdOx1 nCoV-19) vaccines. **Aim**: To investigate the adverse effects of two SARS-CoV-2 vaccines in a real-life preventive setting at the University of Padova. **Methods**: Vaccination was offered to 10116 people. Vaccinated workers were asked to voluntarily report symptoms via online questionnaires sent to them 3 weeks after the first and the second shot. **Results**: 7482 subjects adhered to the vaccination campaign and 6681 subjects were vaccinated with ChAdOx1 nCoV-19 vaccine and 137 (fragile subjects) with the BNT162b2 vaccine. The response rate for both questionnaires was high (i.e., >75%). After the first shot, the ChAdOx1 nCoV-19 vaccine caused more fatigue (*p* < 0.001), headache (*p* < 0.001), myalgia (*p* < 0.001), tingles (*p* = 0.046), fever (*p* < 0.001), chills (*p* < 0.001), and insomnia (*p* = 0.016) than the BNT162b2 vaccine. After the second dose of the BNT162b2 vaccine, more myalgia (*p* = 0.033), tingles (*p* = 0.022), and shivers (*p* < 0.001) than the ChAdOx1 nCoV-19 vaccine were elicited. The side effects were nearly always transient. Severe adverse effects were rare and mostly reported after the first dose of the ChAdOx1 nCoV-19 vaccine. They were dyspnoea (2.3%), blurred vision (2.1%), urticaria (1.3%), and angioedema (0.4%). **Conclusions**: The adverse effects of both vaccines were transient and, overall, mild in severity.

## 1. Introduction

The novel severe acute respiratory syndrome coronavirus 2 (SARS-CoV-2), has caused more than 6.6 million deaths reported to WHO (as of 6th December 2022) and there have been more than 641,000,000 confirmed cases of COVID-19 globally [1].

In this unprecedented pandemic, a quick vaccine development has been considered essential to prevent further morbidity and mortality. Thus, there have been remarkable collaborative worldwide efforts to accelerate preclinical and clinical evaluation of candidate vaccines.

In Italy, according to EMA (European Medicines Agency) indications, the Ministry of Health and the Italian Medicines Agency (Aifa) have given first authorisation to three COVID-19 vaccines: the Pfizer-BioNTech mRNA vaccine (BNT162b2), the Oxford-AstraZeneca adenovirus vectored vaccine (ChAdOx1 nCoV-19), and the Moderna mRNA vaccine (mRNA-1273). At the end of May 2021, the Janssen vaccine (Ad26.COV2-S [recombinant]) was approved too.

On December 2nd 2020, the Italian Minister of Health explained the guidelines of Italy’s Strategic Plan for anti-SARS-CoV-2/COVID-19 vaccination, drafted by the Ministry of Health itself, the Extraordinary Commissioner for the COVID-19 Emergency, the Higher Institute of Health, the Italian National Agency for Regional Healthcare Services (AGENAS), and the Italian Medicines Agency (Aifa) [2].

The plan centred around eight axes, one of them identified the priority categories for vaccination: health and social/medical workers, and residents and staff working in homes for elderly people. In addition, along with the increased availability of vaccines, the category of teachers/professors was identified as an urgent one. The Italian Regions, coordinated by the Ministry of Health and the Extraordinary Commissioner, started the organization of the administration of vaccines.

At mid-February 2021, the head of the health department of the Veneto Region asked the Rector of the University of Padova about the feasibility of an independently organized vaccination campaign. The Rector’s positive response quickly led (on February 26, 2021) to the Veneto Region authorization for starting the COVID-19 vaccination campaign for the University of Padova staff, mainly using the Oxford-AstraZeneca adenovirus vectored vaccine.

Thus, the University of Padova promptly organized the voluntary vaccination of staff, in collaboration with the Preventive Medicine Service, the Departments of the Medical area, the School of Medicine and Surgery, and the Italian Red Cross (IRC, Padova branch) as partners in this initiative. These structures have ensured the necessary medical and nursing staff needed to carry out the campaign.

The aim of this study was to investigate the side effects reported by vaccinated workers of the University of Padova. We used data from subjects who received the ChAdOx1 nCoV-19 vaccine or BNT162b2 vaccine (fragile subjects) between March 2021 and June 2021; symptoms were voluntarily reported via online questionnaires sent to them 3 weeks after the first and the second shot.

## 2. Materials and Methods

On 27 February 2021, COVID-19 vaccination with the ChAdOx1 nCoV-19 vaccine was offered not only to all active workers of Padova University but also to research fellows, PhD students, non-medical postgraduates enrolled in specialization degrees, and collaborators.

This campaign did not involve medical staff, who had already been included in the vaccination initiatives of their respective health facilities. A total of 10116 people was contacted/identified.

Workers and other University affiliates (all these subjects will be named in the text as “workers” due to their work contract at Padova University) were invited to express their willingness to get vaccinated using an online procedure distributed by academic email addresses. Once the survey was concluded, the schedule of the vaccine appointments was determined.

Convocation letters contained strict indications regarding punctuality and compiling a medical history questionnaire at home. This approach was used to allow people enough time to properly complete all fields, avoiding the completion of the questionnaire while queuing for the shots.

Meanwhile, willing participants had the opportunity to write to the e-mail address devoted to the coordination of the vaccination campaign. This was relevant not only for helping participants in logistics or administrative doubts but also in case of concerns on safety and efficacy of the ChAdOx1 nCoV-19 vaccine in relation to previous or current comorbidities of the subjects.

For its estimated higher efficacy (even though derived from clinical trials that excluded, in accordance with the ongoing regulations, immunocompromised subjects, and restricted participation to patients with stable underlying conditions) the BNT162b2 vaccine was recommended by Italian Ministry of Health as more appropriate than ChAdOx1 nCoV-19 vaccine for “fragile patients” or those “with altered immunocompetence” [e.g., affected by severe respiratory, cardiovascular, neurological diseases, diabetes or other severe endocrinopathies, cystic fibrosis, acute or chronic kidney failure, autoimmune disorders, liver disorders, cerebrovascular disorders, oncological disorders, haemoglobinopathy, Down syndrome, solid organ transplant/stem cells transplant, severe obesity (BMI > 35 kg/m^2)].

Thus, the occupational physician derived from the e-mails all cases of workers signalling and documenting their condition of fragility or altered immunocompetence and, according to the availability/accessibility of the BNT162b2 vaccine in the Region, a devoted date was established for these frailest participants. The same date was also set for those who presented for the ChAdOx1 nCoV-19 vaccination at the IRC unit but showed a documented medical history that made them not suitable this vaccine.

The vaccination campaign of the University of Padova started on March 4 and ended on June 26. The patients vaccinated with the BNT162b2 vaccine received a second dose 21 days after the first one; patients vaccinated with the ChAdOx1 nCoV-19 vaccine were contacted 12 weeks after the first shot.

Twenty days after receiving each dose of vaccine, patients were invited by email to fill in a questionnaire on adverse effects. We gave them 5 to 7 days to complete the questionnaire and solicited the voluntary completion by a reminder email describing the helpfulness of these records.

The information about severe adverse reactions was retrieved both from the questionnaire and/or onsite medical data records.

In the questionnaire, participants were asked whether they experienced adverse effects, including both systemic (whole body) and local effects. Solicited systemic adverse effects included: headache, myalgia/arthralgia, blurred vision, numbness, fever, chills, fatigue, lymphadenopathy, dyspnoea, urticaria, diarrhoea, angioedema, diffuse itching, gastro-intestinal disorders (including nausea or vomiting), diarrhoea, faint, oral aphtous stomatitis, insomnia, anosmia, and ageusia.

Solicited local side-effects included local (injection-site) pain, itching, swelling, and redness.

Participants were also permitted to report “no symptoms” by checking a specific box. We also allowed participants to write unstructured/free comments on the adverse effects.

The data were retrieved by the occupational physicians to allow the reporting of the adverse effects to the Ministry of Health and to carry out occupational medical surveillance.

We used a 2-sample test for equality of proportions with Yates continuity correction, which is distributed as a χ2, to compare proportions of self-reporting adverse effects recorded by questionnaires between individuals who received BNT162b2 versus those who received ChAdOx1 nCoV-19. When appropriate, to take into account the multiplicity of tests, we adjusted *p* values with the Holm’s correction. In order to compare self-reporting adverse effects between the first and second doses within the same type of vaccine, we used the McNemar χ2 test for symmetry of rows and columns in a two-dimensional contingency table, which takes into account the dependence within the same units. To make the interpretation of the results clearer, we also plotted the proportions of the different self-reported adverse events versus the time since vaccination at which they appeared.

The analysis was carried out using a statistical software package (R) [3]. A *p* value < 0.05 was considered indicative of statistical significance.

## 3. Results

Out of the 10116 people contacted, 7482 subjects adhered to the vaccination campaign and 6817 went to the vaccination site.

A total of 6681 subjects were vaccinated with ChAdOx1 nCoV-19 vaccine and 137 with the BNT162b2 vaccine (one person received one dose of the ChAdOx1 nCoV-19 vaccine and one dose of the BNT162b2 vaccine).

The number of respondents to the questionnaires that received the ChAdOx1 nCoV-19 vaccine as the first dose was 5794 out 6681 (86.7%); the number of respondents that received the BNT162b2 vaccine was 121 out 121. The number of respondents to the questionnaires who received a second dose of the ChAdOx1 nCoV-19 vaccine was 5095 out 6681; the number of respondents that received the BNT162b2 vaccine was 111 out 121. Thus, the response rate was very high in both groups.

The demographic characteristics of the study population are described in Table 1. Subjects who received the ChAdOx1 nCoV-19 vaccine were younger than those who received the BNT162b2 vaccine. For both vaccines, females and male were equally distributed.

After the first shot of the ChAdOx1 nCoV-19 vaccine, the most common systemic adverse effects were: fatigue (68.5%), fever (60.9%), myalgia/arthralgia (53%), chills (52.5%), and headache (51%). Even though these symptoms were also reported after the second shot, they were significantly less frequently reported (*p* < 0.001). All other investigated systemic symptoms were reported in less than 7% of cases after the first dose and among them, the most frequent was insomnia (6.7%) and the most uncommon was angioedema (0.4%).

After the first shot of the BNT162b2 vaccine, the most common systemic adverse effects were: fatigue (38.02%), headache (26.4%), myalgia and/or arthralgia (16.5%), chills (6.6%), and blurred vision (4.1%). After the second dose of this vaccine the following systemic symptoms were more frequent than after the first dose: fever (18.9% vs. 2.5%), chills (18.9% vs. 6.6%), and insomnia (7.2% vs. 0.8%).

When comparing the systemic side effects of two vaccines after the first shot, the ChAdOx1 nCoV-19 vaccine caused more fatigue (*p* < 0.001), headache (*p* < 0.001), myalgia (*p* < 0.001), tingles (*p* = 0.046), fever (*p* < 0.001), chills (*p* < 0.001), and insomnia (*p* = 0.016) than the BNT162b2 vaccine. On the other hand, after the second dose of the BNT162b2 vaccine, more myalgia (*p* = 0.033), tingles (*p* = 0.022), and shivers (*p* < 0.001) than the ChAdOx1 nCoV-19 vaccine were elicited.

Among the local side effects, the most commonly reported was pain at the injection site with both vaccines that was reported less frequently after the second dose of the ChAdOx1 nCoV-19 vaccine compared to the BNT162b2 vaccine (*p* < 0.001). After the second shot of the ChAdOx1 nCoV-19 vaccine, swelling, redness, and itching were significantly less frequent than after the first dose (*p* < 0.001); this was not the case after the second dose of the BNT162b2 vaccine.

Severe side effects were rare and mostly reported after the first dose of the ChAdOx1 nCoV-19 vaccine. They were: dyspnoea (2.3%), blurred vision (2.1%), urticaria (1.3%), and angioedema (0.4%). No severe side effects were reported after either dose of the BNT162b2 vaccine.

The timing of onset of systemic adverse effects and their duration are described in Figure 1, Figure 2, Figure 3, Figure 4, Figure 5, Figure 6, Figure 7 and Figure 8.

After the first and second dose of the ChAdOx1 nCoV-19 vaccine (Figure 1, Figure 2, Figure 3 and Figure 4), the systemic adverse effects mostly appeared within 6–24 hours. They disappeared within 2 days after the first dose and later (i.e., within 7 days) after the second dose. Diarrhoea and gastrointestinal pain appeared 24 h after the first dose and were the effects that persisted longer after the second dose (3–7 days).

After the first dose of the BNT162b2 vaccine (Figure 5, Figure 6, Figure 7 and Figure 8), the systemic adverse effects mostly appeared within 24 hours and disappeared within 2 days apart from insomnia and gastrointestinal pain, which appeared later (between 24 and 48 h after vaccination), and insomnia that did not disappear within 7 days. After the second dose, the systemic adverse effects appeared earlier (within 24 h after vaccination), apart from diarrhoea and gastrointestinal pain which appeared within 3–7 days. Insomnia and gastrointestinal pain were the most persistent side effects.

Table 2 shows the onsite adverse reactions that occurred within 15 minutes of receiving the first shot of the ChAdOx1 nCoV-19 vaccine and were mostly represented by presyncope (62.6%) and dizziness (20.6%). No severe events were recorded.

More than seven hundred free comments were written, and it was not possible to adequately format them. They were mostly clarifications on symptoms reported in the boxes but in rare cases they described uncommon reports that are summarized in Table 3, Table 4, Table 5 and Table 6 and were all related to subjects who received the ChAdOx1 nCoV-19 vaccine.

## 4. Discussion

The great response of the University of Padova community permitted a successful inception of the vaccination campaign and even though two suspected batches of ChAdOx1 nCoV-19 vaccine were withdrawn during the first days of administrations, the workers and collaborators of our University persevered.

Thus, this is the analysis of a large cohort that provides important information about the occurrence of adverse effects in workers who received two different vaccines. The participants reported a high number of systemic adverse events with both vaccines, but the frequency of severe adverse effects was low.

Comparisons between side effects of the two vaccines should be handled with care due to the relatively low number of BNT162b2 vaccine recipients; however, they can be confidently trusted because they are in line with the results of other relevant studies.

In fact, Menni et al. [4] in a large prospective cohort study in the UK based on data collected by an app, found that systemic adverse events following immunization were reported by 33% of respondents after the first dose of the ChAdOx1 nCoV-19 vaccine and were higher than the number detected after the second dose. For the BNT162b2 vaccine, the number of systemic adverse events following immunization was lower after the first dose (13.5% of participants) and higher after the second dose (22% of participants). A similar trend was confirmed by another study of safety profiles conducted in Australia, Jordan, and the Netherlands [5,6,7].

Our study confirmed that the BNT162b2 vaccine was associated with a lower rate of reactions and the ChAdOx1 nCoV-19 vaccine was associated with higher rates of systemic reactions after the first dose.

These two current emergency authorized COVID-19 vaccines that we used have been evaluated in clinical trials that excluded (in accordance with the ongoing regulations) immunocompromised subjects, and restricted participation to patients with stable underlying conditions (i.e., stable HIV infection) [8,9].

At the time of our campaign the Italian vaccination strategy suggested to use the BNT162b2 vaccine for the so-called “fragile subjects” which is why a small cohort of Padova University workers was selected to receive this vaccine. Thus, someone could argue that the lower exhibition of systemic symptoms could be related to the lower ability to mount an effective immune response in a priori selected patients characterized by having immunosuppressed conditions, such as those induced by immunological disorders or medications.

This might be true, but the cohort included not only immunocompromised patients but also those affected by obesity and severe cardiac, vascular, neurological, hepatic, respiratory disease and the data retrieved did not differ from those documented in clinical trials in which immunocompromised or unstable patients were excluded.

Our population was representative of a diverse population than that selected for clinical trials and made it possible to study the safety profile of two different COVID-19 vaccines in a real-world setting. Considering that our survey was performed after receiving both doses of vaccines, the analysis of adverse effects was more precise than those published last year when the data retrieved by studies were sometimes incomplete because the data on participants receiving the second dose at the time of reporting were missing.

It was a large cohort, younger than those described previously [4,7] and representative of the local community because did not specifically include an a priori selected subgroup (e.g., healthcare workers or elderly people) and last, but not least, included vulnerable subjects.

Considering this final cohort characteristic, different from other cohorts, the assessment of “fragility” was confirmed by the occupational physician and not self-reported by the patients as happened in other previous observational studies. Indeed, thanks to the prompt inception of the vaccination campaign, many fragile workers had the opportunity to receive the vaccine before the vaccination later provided by the National Health Service.

Lastly, we postulate that the time given between the shot and the questionnaire was adequate, not only to monitor adverse effects over time but to also avoid recall bias.

This was a cohort based on subjects who voluntarily agreed to participate in the survey thus someone could argue that this is self-selected cohort and might not represent the general population or that the possibility of missing data on severe reports cannot be excluded.

However, considering the high response rate to the vaccination campaign, the high response rate to the questionnaire and that no workers were excluded from the campaign, we estimate that our setting can be considered a reliable representation of the real-world scenario.

It is likely that some severe adverse effects might have been missed; however, considering that the data were not managed by an app (as in previous studies) but directly by the occupational physicians in charge of all subjects, we think that it is unlikely that severe events might have been missed. This also because subjects had the possibility to communicate with doctors (by phone or e-mail) in every moment of the campaign, including after the administration of the questionnaires.

Another limitation is that we evaluated only short-term adverse effects. Almost all side effects disappeared within 7 days but, as shown by Figure 8, insomnia and gastrointestinal pain were ongoing (within 3–7 days) after the second dose of the BNT162b2 vaccine, thus we cannot say anything on the timing of their remission.

## 5. Conclusions

The context of motivated collaboration among key sectors of society and the academic community made a rapid and successful vaccination campaign possible.

The adverse effects of both vaccines were transient and, overall, mild in severity; thus, the safety of the vaccines described is reassuring, and no unexpected patterns of concern were detected.

## Figures and Tables

**Figure 1 vaccines-11-00951-f001:**
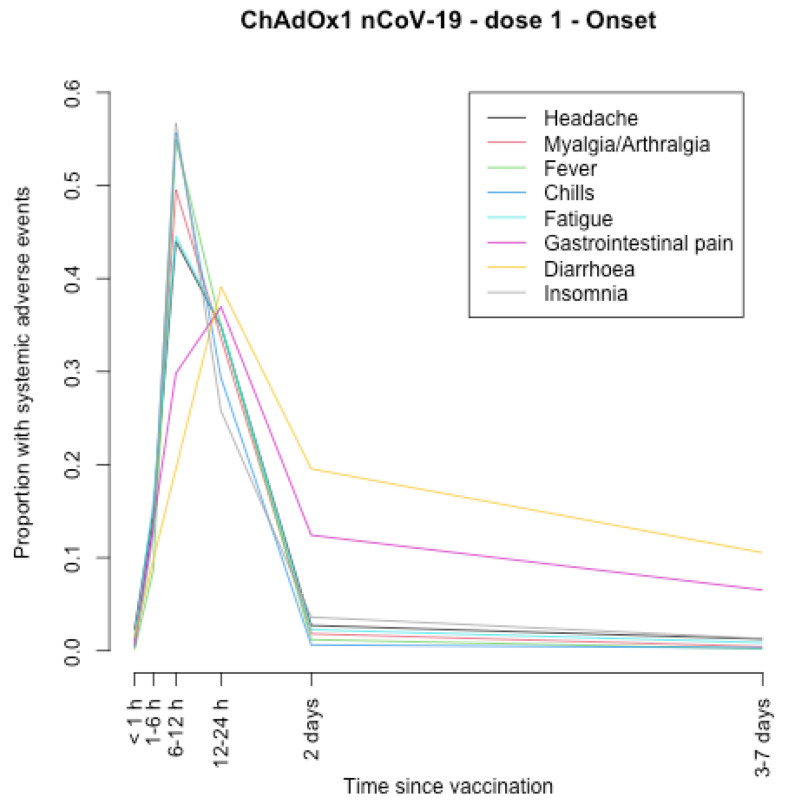
Timing of onset of systemic adverse effects of ChAdOx1 nCoV-19 vaccine after the first dose.

**Figure 2 vaccines-11-00951-f002:**
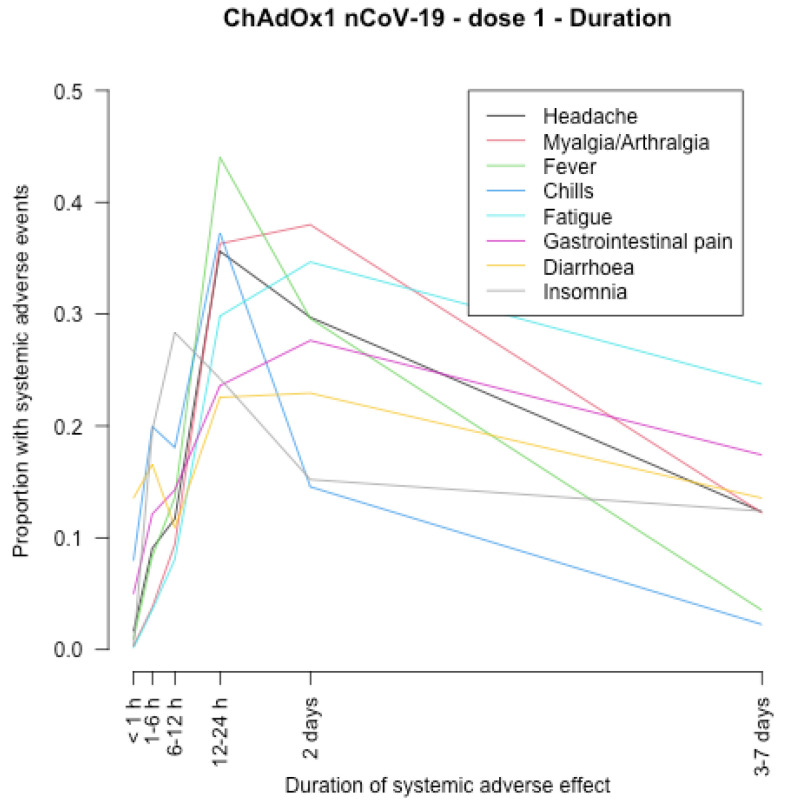
Duration of systemic adverse effects of ChAdOx1 nCoV-19 vaccine after the first dose.

**Figure 3 vaccines-11-00951-f003:**
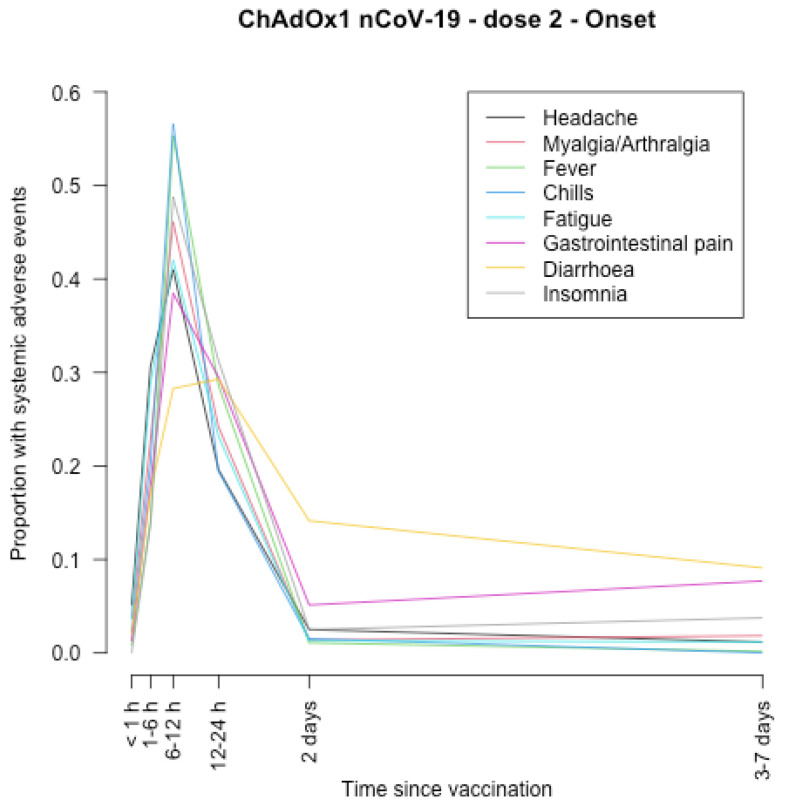
Timing of onset of systemic adverse effects of ChAdOx1 nCoV-19 vaccine after the second dose.

**Figure 4 vaccines-11-00951-f004:**
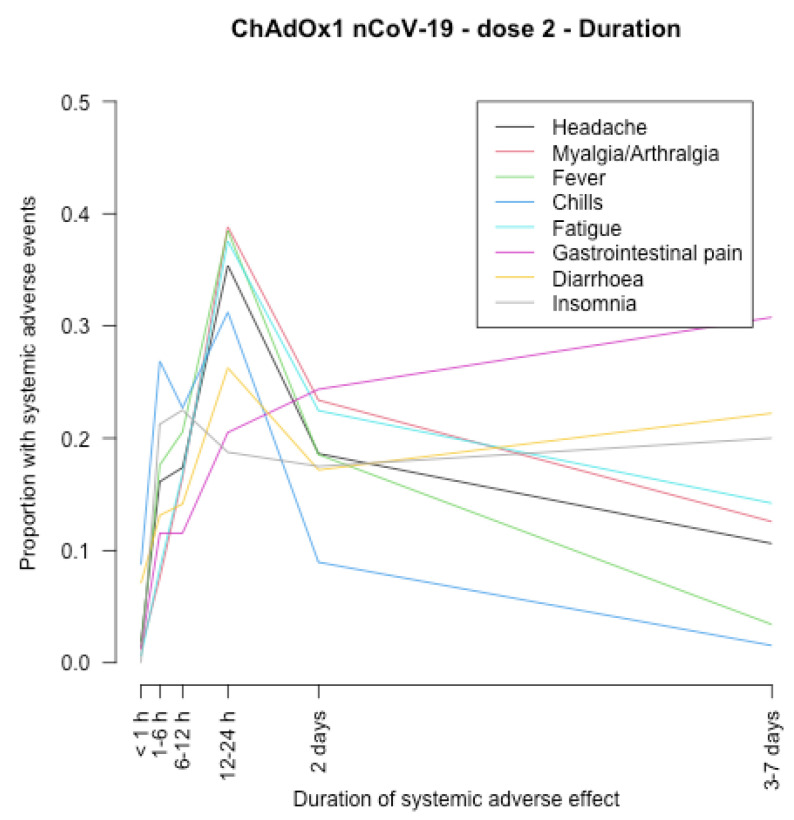
Duration of systemic adverse effects of ChAdOx1 nCoV-19 vaccine after the second dose.

**Figure 5 vaccines-11-00951-f005:**
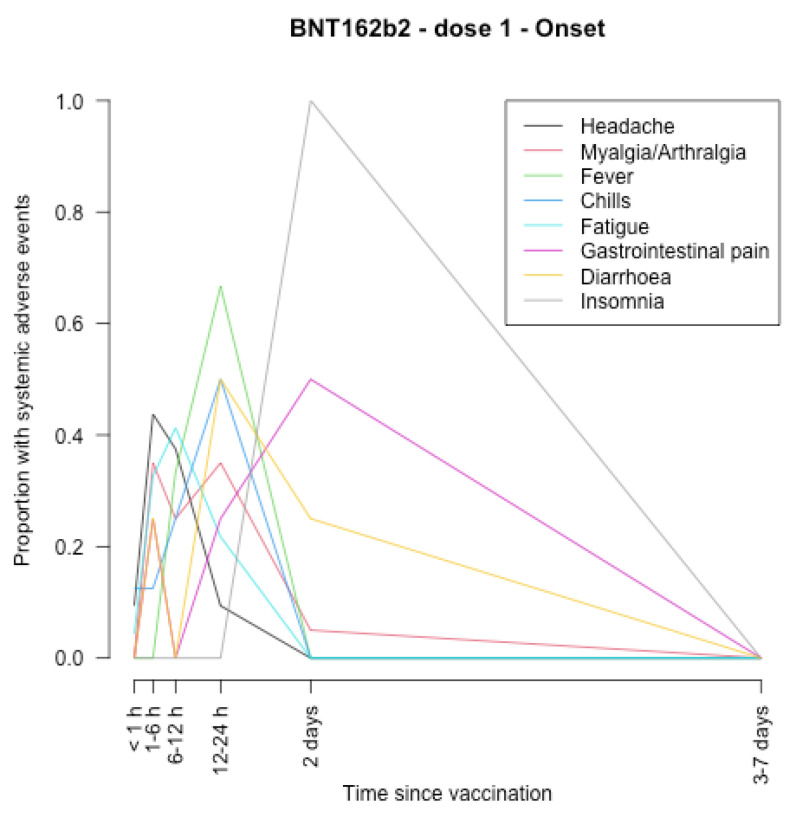
Timing of onset of systemic adverse effects of BNT162b2 vaccine after the first dose.

**Figure 6 vaccines-11-00951-f006:**
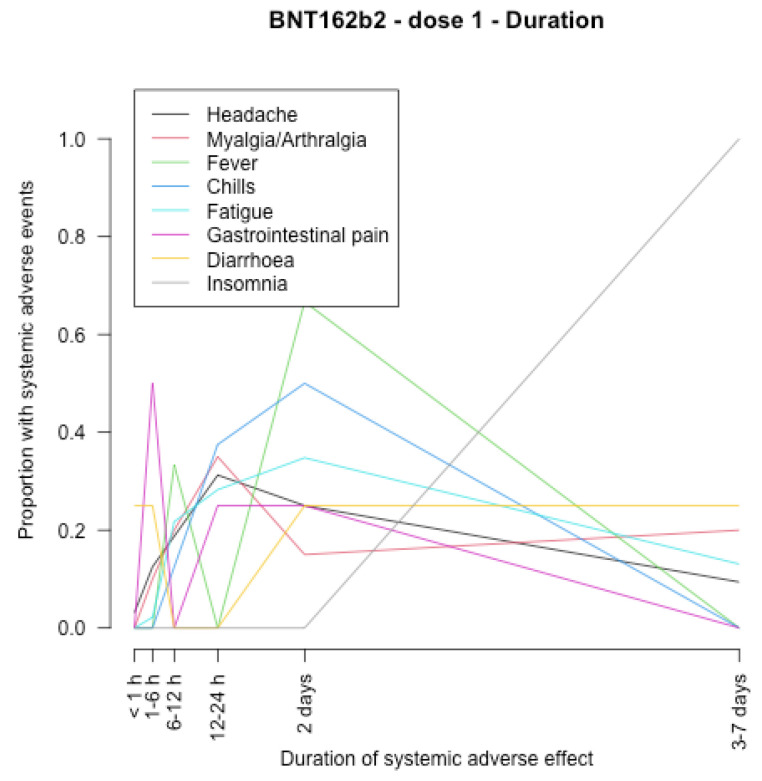
Duration of systemic adverse effects of BNT162b2 vaccine after the first dose.

**Figure 7 vaccines-11-00951-f007:**
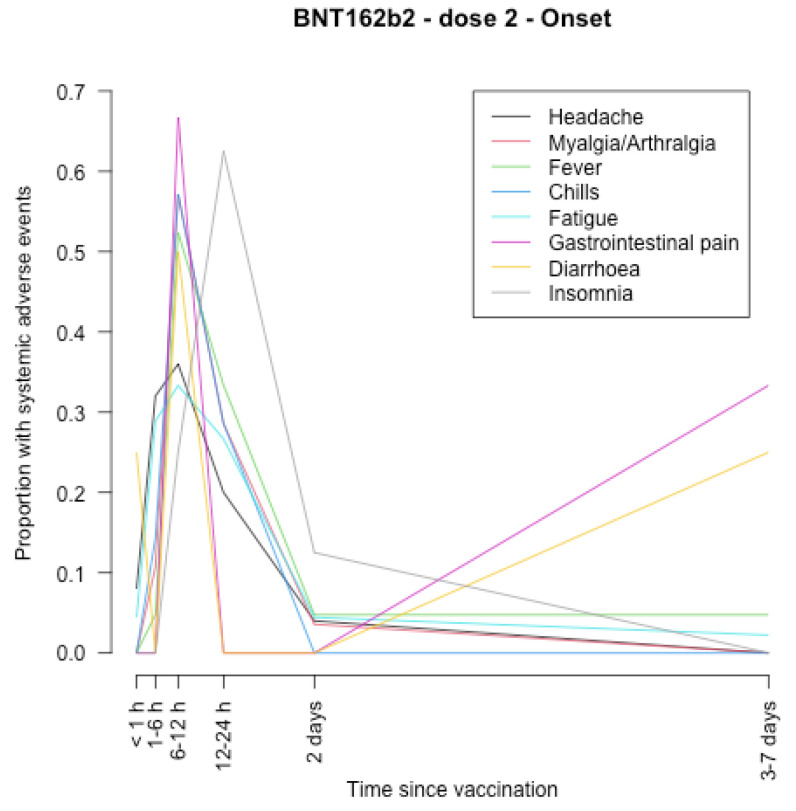
Timing of onset of systemic adverse effects of BNT162b2 vaccine after the second dose.

**Figure 8 vaccines-11-00951-f008:**
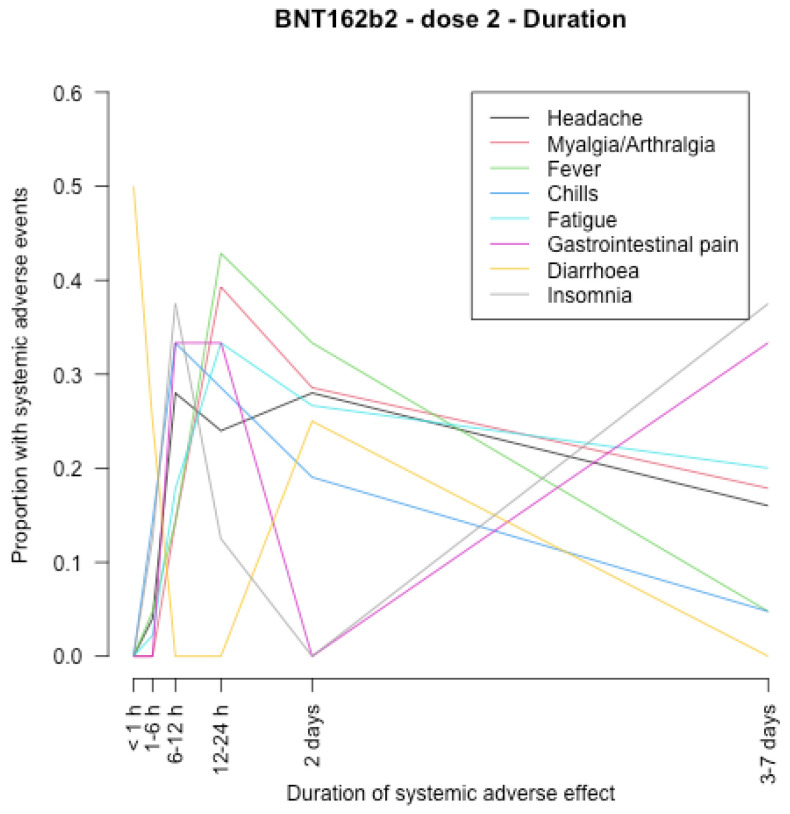
Duration of systemic adverse effects of BNT162b2 vaccine after the second dose.

**Table 1 vaccines-11-00951-t001:** Demographic characteristics of the study population and participants self-reporting adverse effects recorded by questionnaires sent by email 20 days after vaccinations.

	ChAdOx1 nCoV-19 Vaccine			BNT162b2 Vaccine			ChAdOx1 nCoV-19 Vaccine vs. BNT162b2 Vaccine	
	First Dose	Second Dose	First Dose vs. Second Dose	First Dose	Second Dose	First Dose vs. Second Dose (*p* Value)	First Dose (*p* Value)	Second Dose (*p* Value)
Gender M F	3062 (52.8%) 2732 (47.2%)	2654 (52.1%) 2441 (47.9%)		70 (57.9%) 51 (42.1%)	57 (51.4%) 54 (48.6%)			
Age 18–59 ≥ 60	5139 (88.7%) 655 (11.3%)	4485 (88%) 610 (12%)		86 (71.1%) 35 (28.9%)	84 (75.7%) 27 (24.3%)			
Systemic Adverse Effects								
Headache	2982 (51%)	1122 (22%)	<0.001	32 (26.4%)	25 (22.5%)	0.589	<0.001	0.992
Myalgia/ Arthralgia	3069 (53%)	868 (17%)	<0.001	20 (16.5%)	28 (25.3%)	0.141	<0.001	0.033
Blurred Vision	123 (2.1%)	54 (1.1%)	<0.001	5 (4.1%)	1 (0.9%)	0.256	0.239	1
Tingles (at least 1) Head Arms Thorax Abdomen Legs								
	448 (7.7%)	151 (3%)	<0.001	3 (2.5%)	8 (7.2%)	0.166	0.046	0.022
	76 (1.3%)	28 (0.5%)	<0.001	2 (1.7%)	1 (0.09%)	1	1	1
	257 (4.4%)	82 (1.6%)	<0.001	1 (0.8%)	8 (7.2%)	0.03	0.088	<0.001
	35 (0.6%)	9 (0.2%)	<0.001	0 (0%)	1 (0.9%)	1	0.793	0.530
	17 (0.3%)	5 (0.1%)	0.039	1 (0.8%)	0 (0%)	1	0.829	1
	199 (3.4%)	76 (1.5%)	<0.001	0 (0%)	1 (0.9%)	1	0.0680	0.910
Fever (>37.5)	3526 (60.9%)	647 (12.7%)	<0.001	3 (2.5%)	21 (18.9%)	<0.001	<0.001	0.073
Chills	3044 (52.5%)	458 (9%)	<0.001	8 (6.6%)	21 (18.9%)	0.001	<0.001	0.001
Fatigue	3973 (68.5%)	1986 (39.0%)	<0.001	46 (38.02%)	45 (40.54%)	0.796	<0.001	0.081
Swollen Armpit	206 (3.6%)	67 (1.3%)	<0.001	4 (3.3%)	5 (4.5%)	0.895	1	0.015
Dyspnoea	132 (2.3%)	34 (0.7%)	<0.001	1 (0.8%)	1 (0.9%)	1	0.448	1
Urticaria (at least 1) Head Arms Thorax Abdomen Legs								
	75 (1.3%)	31 (0.6%)	<0.001	1 (9.8%)	0 (0%)		0.959	0.841
	15 (0.3%)	7 (0.1%)	0.227	0 (0%)	0 (0%)	1	1	1
	26 (0.4%)	18 (0.4%)	0.516	0 (0%)	0 (0%)		0.962	0.942
	17 (0.3%)	5 (0.1%)	0.039	1 (0.8%)	0 (0%)		0.829	1
	18 (0.3%)	10 (0.2%)	0.317	0 (0%)	0 (0%)	1	1	1
	32 (0.6%)	12 (0.2%)	0.014	0 (0%)	0 (0%)		0.844	
Angioedema (at least 1) Head Arms Thorax Abdomen Legs								
	25 (0.4%)	4 (0.08%)	0.001	1 (0.08%)	0 (0%)	1	1	1
	10 (0.2%)	1 (0.02%)	0.027	0 (0%)	0 (0%)		1	1
	6 (0.1%)	0 (0%)	0.058	0 (0%)	0 (0%)		1	1
	7 (0.1%)	1 (0.02%)	0.110	0 (0%)	0 (0%)		1	1
	10 (0.2%)	2 (0.04%)	0.070	1 (0.08%)	0 (0%)	1	0.560	1
Diffuse Itching	61 (1.1%)	27 (0.5%)	0.003	0 (0%)	1 (0.09%)	0.965	0.494	1
Gastrointestinal Pain (Including Nausea)	322 (5.6%)	78 (1.5%)	<0.001	4 (3.3%)	3 (2.7%)	1	0.376	0.549
Diarrhoea	266 (4.6%)	99 (1.9%)	<0.001	4 (3.3%)	4 (3.6%)	1	0.644	0.369
Faint	84 (1.4%)	10 (0.2%)	<0.001	1 (0.08%)	1 (0.09%)	1	0.848	0.579
Oral Aphtous	75 (1.3%)	15 (6.8%)	<0.001	0 (0%)	1 (0.09%)	0.965	0.393	0.376
Insomnia	388 (6.7%)	80 (1.6%)	<0.001	1 (0.08%)	8 (7.2%)	0.030	0.016	<0.001
Anosmia	16 (0.3%)	3 (0.1%)	0.013	0 (0%)	0 (0%)	1	1	1
Ageusia	54 (0.9%)	5 (0.1%)	<0.001	0 (0%)	0 (0%)	1	0.557	0.537
Local Adverse Effects								
Pain	4311 (74.4%)	2644 (51.9%)	<0.001	95 (78.5%)	76 (68.5%)	0.113	0.408	0.001
Itching	320 (5.5%)	185 (3.6%)	<0.001	8 (6.6%)	3 (2.7%)	0.276	0.762	0.794
Swelling	566 (9.8%)	286 (5.6%)	<0.001	13 (10.7%)	10 (9%)	0.824	0.854	0.186
Redness	373 (6.4%)	222 (4.4%)	<0.001	8 (6.6%)	5 (4.5%)	0.680	1	1

**Table 2 vaccines-11-00951-t002:** Onsite adverse reactions within 15 minutes of receiving the first shot of ChAdOx1 nCoV-19 vaccine.

Total Onsite Adverse Reactions	N°
Presyncope	29
Dizziness	14
Syncope	5
Swallowing difficulties	2
Asthenia	2
Hypertension	2
Tachycardia	2
Nausea	2
Palpitations	2
Headache	2
Dyspnoea	2
Erythema	1
Burning chest pain	1
Hot flash	1
Hypotension	1
**Total**	**68**

**Table 3 vaccines-11-00951-t003:** Cardiovascular and haematological symptoms reported after receiving ChAdOx1 nCoV-19 vaccine.

Cardiovascular and Haematological Symptoms	1st Dose	2nd Dose
Tachycardia	41	6
Hypertension	11	3
Palpitations	10	2
Bleeding	10	7
Hypotension	6	3
Haematomas	5	4
Arrhythmia	4	2
Thrombocytosis	2	1

**Table 4 vaccines-11-00951-t004:** Neurovegetative symptoms reported after receiving ChAdOx1 nCoV-19 vaccine.

Neurovegetative Symptoms	1st Dose	2nd Dose
Dizziness	65	15
Anorexia	24	1
Asthenia	21	8
Altered thermoregulation	21	2
Confusion or difficulty concentrating	14	11
Excessive sweating	13	4
Drowsiness	12	5
Hyperphagia	6	1
Polydipsia	4	0
Anxiety, >12 h	2	0
Transient global amnesia	2	0
Tinnitus	2	1
Seizures	0	1
Transient ischemic attack	0	1

**Table 5 vaccines-11-00951-t005:** Pain symptoms reported after receiving ChAdOx1 nCoV-19 vaccine.

Pain	1st Dose	2nd Dose
Back Pain	25	2
Earache	8	1
Neck pain	6	0
Odontalgia	5	0
Thoracic pain (not cardiac)	5	2

**Table 6 vaccines-11-00951-t006:** Mixed symptoms reported after receiving ChAdOx1 nCoV-19 vaccine.

Miscellaneous	1st Dose	2nd Dose
Ocular discomfort	25	2
Cough	14	2
Hoarse voice or laryngitis	10	0
Sour taste in the mouth	6	0
Amenorrhea	4	6
Oral herpes zoster	3	3
Flu-like symptoms	3	4
Varicocele	2	0
Haemorrhoids	1	1
Herpes virus infection	1	1
Yellow and mildly swollen tongue	1	0
Appendicitis	1	0
Constipation	1	0
Dysgeusia	1	0
Pytiriasis rosea	1	2
Parosmia	0	1
Psoriasis	0	1
Alopecia areata	0	1

## Data Availability

Appropriate forms of data sharing could be arranged after a reasonable request to the first author.

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
