# Peer review of "Comparison of Adverse Effects of Two SARS-CoV-2 Vaccines Administered in Workers of the University of Padova"

_vaccines, 2023, doi:10.3390/vaccines11050951_

Round 1

Reviewer 1 Report

This is an interesting manuscript dealing with vaccination against COVID-19 pandemic coming from a single center university center. Although there is no striking novelty, the manuscript portends some interesting informations. I would like to suggest some changes to the paper, and to offer the opportunity to authors to discuss some obscure points to me: 1) I think that Statistical Analysis should be highlighted in the Methods section; 2) Some explanation might be given to the high frequency of, and the long duration of, insomnia following vaccination?; 3) In the authors view, what is the explanation for the low prevalence of thrombosis, since the majority ofmpatients were vaccinated with Astra Zeneca vaccine? 4) Conclusion: the authors claim that the vaccination campaign was succesful. However, about 33% of the invited people declined the invitation. If we take into account that the campaign was made in a literate and high-income region, this seems to be a very high rate of declination. Some comments on this point might be desirable.     

Author Response

We thank the Reviewer for the overall positive comments.

1) We took the hint and add details in the “Methods” section.

2) We have hypothesized that insomnia was related to a sort of anxiety and mass-media pressure related to these new vaccines and their mostly unknown adverse effects.

3) We cannot give a precise response to this question but probably the personal history taken by occupational physicians permitted to define more properly if patients were at risk for thrombosis or not, and thus fragile subjects (from immune-haematological point of view) were directly suggested to get vaccinated with BNT162b2 vaccine.

4) We wrote that the vaccination campaign was successful because, nevertheless the media-pressure on  adverse effects of AstraZeneca vaccine and the opportunity that some patients have had to get the vaccination offered by local health facilities,  a consistent number still adhered to the campaign.

Reviewer 2 Report

I would just let the authors of this manuscript: MY CONGRATULATIONS!! It has been done a really scientific work and survey during the caotic pandemic situation and the resulting outputs are also scientifically analysed and nicely reported in this manuscript.

Author Response

We warmly thank the Reviewer for the appreciation.

Reviewer 3 Report

This interesting prospective cohort study measures the adverse effects of two SARS-CoV-2 vaccines administered in workers of the University of Padova.

The introduction is clear and presents the rationale. However, the conclusion states that the adverse effects of both vaccines were moderate in frequency but there is no mention of frequency in the results. It would be interesting to add frequencies of the most common adverse effects in the results.

In the materials and methods section, the population, location, study inclusion period, follow-up and data collection are well defined. In the statistical part, χ2 tests are used to compare individuals who received BNT162b2 versus those who received ChAdOx1 nCoV-19 and between individuals who received first and second dose within the same type of vaccine. For these second comparisons, McNemar χ2 tests must be used to take into account the absence of independence.

In the results section, the flow chart is described. I do not understand why line 142 there are 137 subjects with the BNT162b2 vaccine and only 121 on lines 147 and 149. The frequencies of the most common adverse effects are not accompanied by confidence intervals. It would be interesting to add the confidence intervals. The frequency and its confidence interval of the absence of adverse effect could be added.

The discussion part is clear and well written.

In the conclusion, it does not seem possible to me to say that the adverse effects are moderate in frequency when almost 70% of the subjects reported fatigue and 61% of the subjects reported fever.

Finally, there is one typo: 

line 104, square meter is written m^2 instead of m2.

Author Response

This interesting prospective cohort study measures the adverse effects of two SARS-CoV-2 vaccines administered in workers of the University of Padova.

Thanks for the appreciation.

The introduction is clear and presents the rationale. However, the conclusion states that the adverse effects of both vaccines were moderate in frequency but there is no mention of frequency in the results. It would be interesting to add frequencies of the most common adverse effects in the results.

Thanks for the comment, we amended the text. The frequencies are reported between brackets in the Results section and in the Tables.

In the materials and methods section, the population, location, study inclusion period, follow-up and data collection are well defined. In the statistical part, χ2 tests are used to compare individuals who received BNT162b2 versus those who received ChAdOx1 nCoV-19 and between individuals who received first and second dose within the same type of vaccine. For these second comparisons, McNemar χ2 tests must be used to take into account the absence of independence.

We thank the Reviewer for the suggestion. As suggested, for the comparison between first and second dose, we now used the McNemar test to take into account the dependence within the same units. We also include this method in the Materials and Methods section.

In the results section, the flow chart is described. I do not understand why line 142 there are 137 subjects with the BNT162b2 vaccine and only 121 on lines 147 and 149. The frequencies of the most common adverse effects are not accompanied by confidence intervals. It would be interesting to add the confidence intervals. The frequency and its confidence interval of the absence of adverse effect could be added.

Thanks for the suggestion but including the confidence intervals seems us too difficult to interpret due to too many numbers. Anyway we added here an example of possible Table to explain this abundance of data.

The discussion part is clear and well written.

In the conclusion, it does not seem possible to me to say that the adverse effects are moderate in frequency when almost 70% of the subjects reported fatigue and 61% of the subjects reported fever.

Thanks for the comment, we amended the text.

Finally, there is one typo:

line 104, square meter is written m^2 instead of m2.

Thanks, we have corrected the typo.

ChAdOx1nCoV-19vaccine

BNT162b2vaccine

Firstdose

Seconddose

Firstdose

Seconddose

est%

lwr.ci%

upr.ci%

est%

lwr.ci%

upr.ci%

est%

lwr.ci%

upr.ci%

est%

lwr.ci%

upr.ci

GENDER

M

52.8%

51.6%

54.1%

52.1%

50.7%

53.5%

57.9%

48.5%

66.8%

51.4%

41.7%

61.0

F

47.2%

45.9%

48.4%

47.9%

46.5%

49.3%

42.1%

33.2%

51.5%

48.6%

39.0%

58.3

AGE18-59

88.7%

87.9%

89.5%

88.0%

87.1%

88.9%

71.1%

62.1%

79.0%

75.7%

66.6%

83.3

AGE≥60

11.3%

10.5%

12.1%

12.0%

11.1%

12.9%

28.9%

21.0%

37.9%

24.3%

16.7%

33.4

HEADACHE

51.5%

50.2%

52.8%

22.0%

20.9%

23.2%

26.4%

18.8%

35.2%

22.5%

15.1%

31.4

MYALGIA/ARTHRALGIA

53.0%

51.7%

54.3%

17.0%

16.0%

18.1%

16.5%

10.4%

24.4%

25.2%

17.5%

34.4

BLURRED VISION

2.1%

1.8%

2.5%

1.1%

0.8%

1.4%

4.1%

1.4%

9.4%

0.9%

0.0%

4.9

TINGLES (ATLEAST1)

7.7%

7.1%

8.5%

3.0%

2.5%

3.5%

2.5%

0.5%

7.1%

7.2%

3.2%

13.7

TINGLES HEAD

1.3%

1.0%

1.6%

0.5%

0.4%

0.8%

1.7%

0.2%

5.8%

0.9%

0.0%

4.9

TINGLES ARMS

4.4%

3.9%

5.0%

1.6%

1.3%

2.0%

0.8%

0.0%

4.5%

7.2%

3.2%

13.7

TINGLES THORAX

0.6%

0.4%

0.8%

0.2%

0.1%

0.3%

0.0%

0.0%

3.0%

0.9%

0.0%

4.9

TINGLES ABDOMEN

0.3%

0.2%

0.5%

0.1%

0.0%

0.2%

0.8%

0.0%

4.5%

0.0%

0.0%

3.3

TINGLES LEGS

3.4%

3.0%

3.9%

1.5%

1.2%

1.9%

0.0%

0.0%

3.0%

0.9%

0.0%

4.9

FEVER

60.9%

59.6%

62.1%

12.7%

11.8%

13.6%

2.5%

0.5%

7.1%

18.9%

12.1%

27.5

CHILLS

52.5%

51.2%

53.8%

9.0%

8.2%

9.8%

6.6%

2.9%

12.6%

18.9%

12.1%

27.5

FATIGUE

68.6%

67.4%

69.8%

39.0%

37.6%

40.3%

38.0%

29.3%

47.3%

40.5%

31.3%

50.3

SWOLLEN ARMPIT

3.6%

3.1%

4.1%

1.3%

1.0%

1.7%

3.3%

0.9%

8.2%

4.5%

1.5%

10.2

DYSPNOEA

2.3%

1.9%

2.7%

0.7%

0.5%

0.9%

0.8%

0.0%

4.5%

0.9%

0.0%

4.9

URTICARIA (ATLEAST1)

1.3%

1.0%

1.6%

0.6%

0.4%

0.9%

0.8%

0.0%

4.5%

0.0%

0.0%

3.3

URTICARIA HEAD

0.3%

0.1%

0.4%

0.1%

0.1%

0.3%

0.0%

0.0%

3.0%

0.0%

0.0%

3.3

URTICARIA ARMS

0.4%

0.3%

0.7%

0.4%

0.2%

0.6%

0.0%

0.0%

3.0%

0.0%

0.0%

3.3

URTICARIA THORAX

0.3%

0.2%

0.5%

0.1%

0.0%

0.2%

0.8%

0.0%

4.5%

0.0%

0.0%

3.3

URTICARIA ABDOMEN

0.3%

0.2%

0.5%

0.2%

0.1%

0.4%

0.0%

0.0%

3.0%

0.0%

0.0%

3.3

URTICARIA LEGS

0.6%

0.4%

0.8%

0.2%

0.1%

0.4%

0.0%

0.0%

3.0%

0.0%

0.0%

3.3

ANGIOEDEMA (ATLEAST1)

0.4%

0.3%

0.6%

0.1%

0.0%

0.2%

0.8%

0.0%

4.5%

0.0%

0.0%

3.3

ANGIOEDEMA HEAD

0.2%

0.1%

0.3%

0.0%

0.0%

0.1%

0.0%

0.0%

3.0%

0.0%

0.0%

3.3

ANGIOEDEMA ARMS

0.1%

0.0%

0.2%

0.0%

0.0%

0.1%

0.0%

0.0%

3.0%

0.0%

0.0%

3.3

ANGIOEDEMA THORAX

0.1%

0.0%

0.2%

0.0%

0.0%

0.1%

0.0%

0.0%

3.0%

0.0%

0.0%

3.3

ANGIOEDEMA ABDOMEN

0.2%

0.1%

0.3%

0.0%

0.0%

0.1%

0.8%

0.0%

4.5%

0.0%

0.0%

3.3

DIFFUSE ITCHING

1.1%

0.8%

1.4%

0.5%

0.3%

0.8%

0.0%

0.0%

3.0%

0.9%

0.0%

4.9

GASTROINTESTINAL PAIN

5.6%

5.0%

6.2%

1.5%

1.2%

1.9%

3.3%

0.9%

8.2%

2.7%

0.6%

7.7

DIARRHOEA

4.6%

4.1%

5.2%

1.9%

1.6%

2.4%

3.3%

0.9%

8.2%

3.6%

1.0%

9.0

FAINT

1.4%

1.2%

1.8%

0.2%

0.1%

0.4%

0.8%

0.0%

4.5%

0.9%

0.0%

4.9

ORAL APHTOUS

1.3%

1.0%

1.6%

0.3%

0.2%

0.5%

0.0%

0.0%

3.0%

0.9%

0.0%

4.9

INSOMNIA

6.7%

6.1%

7.4%

1.6%

1.2%

2.0%

0.8%

0.0%

4.5%

7.2%

3.2%

13.7

ANOSMIA

0.3%

0.2%

0.4%

0.1%

0.0%

0.2%

0.0%

0.0%

3.0%

0.0%

0.0%

3.3

AGEUSIA

0.9%

0.7%

1.2%

0.1%

0.0%

0.2%

0.0%

0.0%

3.0%

0.0%

0.0%

3.3

PAIN

74.4%

73.3%

75.5%

51.9%

50.5%

53.3%

78.5%

70.1%

85.5%

85.6%

77.6%

91.5

ITCHING

5.5%

4.9%

6.1%

3.6%

3.1%

4.2%

6.6%

2.9%

12.6%

7.2%

3.2%

13.7

SWELLING

9.8%

9.0%

10.6%

5.6%

5.0%

6.3%

10.7%

5.8%

17.7%

11.7%

6.4%

19.2

REDNESS

6.4%

5.8%

7.1%

4.4%

3.8%

5.0%

6.6%

2.9%

12.6%

7.2%

3.2%

13.7

Reviewer 4 Report

1.       It is not clear from what is presented why the ‘workers’ at the University of Padova are special in that they should be examined separately from the general population, students, and/or other adults in the community/country, or world. Is there some factor that would make one hypothesize this university population would be different from other populations in regard to responses to one or more of the COVID vaccines examined? This is not addressed in the manuscript. The Discussion section mentions other studies of adverse events to COVID vaccines but does not directly compare/discuss the results of these other studies with the results of the present study.

2.       The title states the subjects of the experiment were ‘workers’ at the University of Padova. However, the Abstract states ‘teachers’ at the university were the subjects of investigation. The Methods section states ‘all active workers’ and university affiliates except medical staff were the subjects of the study. The Methods section also indicates that Ph.D. students, research fellows, and others were invited to participate (employees are not the same as students?). As a result, there is confusion as to the actual subjects of the study. The actual subjects of the study should be clarified.

3.       The data collection was ‘voluntary’. This suggests there may be some participation bias in the study. This should be addressed. Relatedly, the reader is not told if the subjects for the study received compensation for participation in the study. Did the subjects supervisors know about the participation in the study? Was participation in any way linked to performance reviews or some other employment-related factor?

4.       The readers are not told what types of post hoc analyses were used after the omnibus chi-square tests to determine which groups different from each other. Where standardized residuals calculated and used? Was a Bonferroni correction to alpha used? Perhaps are more detailed appraisal of what independent variables were used in the analyses would be helpful.

5.       The authors should consider the inclusion of confidence intervals (maybe 95% confidence intervals) in Table 1. This would be helpful in understanding the characteristics of the side effects examined. This information might be more useful in interpreting the systemic adverse effects reported that the p-values populating the table as the methods used to generate these p-values are not clear.

6.       The number of significant digits in Table 1 and throughout the paper are not constant. This creates unnecessary confusion.

7.       Figures 1 through 8 are unnecessarily complicated given the point the authors are trying to make and the stated purpose of the study. It appears the authors want the reader to learn something about the onset and duration of side effects BETWEEN the different vaccines after the first and second doses of each. If this is the case, the figures presented in the manuscript should capture these differences. Perhaps the authors can calculate a way to capture DIFFERENCES in the onset or duration of the side-effects BETWEEN the vaccines? At present, the only way to discern what the authors state is the purpose of the research is to flip awkwardly between figures.

Tables 2 through 6 suffer from the same disconnect as Figures 1 through 8. The stated purpose of the study is to compare side effects from different vaccines. However, the tables only one vaccine are presented. This makes a comparison BETWEEN vaccines difficult for the reader to understand. Perhaps there is a better way to provide context for these tables?

Author Response

ChAdOx1nCoV-19vaccine

BNT162b2vaccine

Firstdose

Seconddose

Firstdose

Seconddose

est%

lwr.ci%

upr.ci%

est%

lwr.ci%

upr.ci%

est%

lwr.ci%

upr.ci%

est%

lwr.ci%

upr.ci

GENDER

M

52.8%

51.6%

54.1%

52.1%

50.7%

53.5%

57.9%

48.5%

66.8%

51.4%

41.7%

61.0

F

47.2%

45.9%

48.4%

47.9%

46.5%

49.3%

42.1%

33.2%

51.5%

48.6%

39.0%

58.3

AGE18-59

88.7%

87.9%

89.5%

88.0%

87.1%

88.9%

71.1%

62.1%

79.0%

75.7%

66.6%

83.3

AGE≥60

11.3%

10.5%

12.1%

12.0%

11.1%

12.9%

28.9%

21.0%

37.9%

24.3%

16.7%

33.4

HEADACHE

51.5%

50.2%

52.8%

22.0%

20.9%

23.2%

26.4%

18.8%

35.2%

22.5%

15.1%

31.4

MYALGIA/ARTHRALGIA

53.0%

51.7%

54.3%

17.0%

16.0%

18.1%

16.5%

10.4%

24.4%

25.2%

17.5%

34.4

BLURRED VISION

2.1%

1.8%

2.5%

1.1%

0.8%

1.4%

4.1%

1.4%

9.4%

0.9%

0.0%

4.9

TINGLES (ATLEAST1)

7.7%

7.1%

8.5%

3.0%

2.5%

3.5%

2.5%

0.5%

7.1%

7.2%

3.2%

13.7

TINGLES HEAD

1.3%

1.0%

1.6%

0.5%

0.4%

0.8%

1.7%

0.2%

5.8%

0.9%

0.0%

4.9

TINGLES ARMS

4.4%

3.9%

5.0%

1.6%

1.3%

2.0%

0.8%

0.0%

4.5%

7.2%

3.2%

13.7

TINGLES THORAX

0.6%

0.4%

0.8%

0.2%

0.1%

0.3%

0.0%

0.0%

3.0%

0.9%

0.0%

4.9

TINGLES ABDOMEN

0.3%

0.2%

0.5%

0.1%

0.0%

0.2%

0.8%

0.0%

4.5%

0.0%

0.0%

3.3

TINGLES LEGS

3.4%

3.0%

3.9%

1.5%

1.2%

1.9%

0.0%

0.0%

3.0%

0.9%

0.0%

4.9

FEVER

60.9%

59.6%

62.1%

12.7%

11.8%

13.6%

2.5%

0.5%

7.1%

18.9%

12.1%

27.5

CHILLS

52.5%

51.2%

53.8%

9.0%

8.2%

9.8%

6.6%

2.9%

12.6%

18.9%

12.1%

27.5

FATIGUE

68.6%

67.4%

69.8%

39.0%

37.6%

40.3%

38.0%

29.3%

47.3%

40.5%

31.3%

50.3

SWOLLEN ARMPIT

3.6%

3.1%

4.1%

1.3%

1.0%

1.7%

3.3%

0.9%

8.2%

4.5%

1.5%

10.2

DYSPNOEA

2.3%

1.9%

2.7%

0.7%

0.5%

0.9%

0.8%

0.0%

4.5%

0.9%

0.0%

4.9

URTICARIA (ATLEAST1)

1.3%

1.0%

1.6%

0.6%

0.4%

0.9%

0.8%

0.0%

4.5%

0.0%

0.0%

3.3

URTICARIA HEAD

0.3%

0.1%

0.4%

0.1%

0.1%

0.3%

0.0%

0.0%

3.0%

0.0%

0.0%

3.3

URTICARIA ARMS

0.4%

0.3%

0.7%

0.4%

0.2%

0.6%

0.0%

0.0%

3.0%

0.0%

0.0%

3.3

URTICARIA THORAX

0.3%

0.2%

0.5%

0.1%

0.0%

0.2%

0.8%

0.0%

4.5%

0.0%

0.0%

3.3

URTICARIA ABDOMEN

0.3%

0.2%

0.5%

0.2%

0.1%

0.4%

0.0%

0.0%

3.0%

0.0%

0.0%

3.3

URTICARIA LEGS

0.6%

0.4%

0.8%

0.2%

0.1%

0.4%

0.0%

0.0%

3.0%

0.0%

0.0%

3.3

ANGIOEDEMA (ATLEAST1)

0.4%

0.3%

0.6%

0.1%

0.0%

0.2%

0.8%

0.0%

4.5%

0.0%

0.0%

3.3

ANGIOEDEMA HEAD

0.2%

0.1%

0.3%

0.0%

0.0%

0.1%

0.0%

0.0%

3.0%

0.0%

0.0%

3.3

ANGIOEDEMA ARMS

0.1%

0.0%

0.2%

0.0%

0.0%

0.1%

0.0%

0.0%

3.0%

0.0%

0.0%

3.3

ANGIOEDEMA THORAX

0.1%

0.0%

0.2%

0.0%

0.0%

0.1%

0.0%

0.0%

3.0%

0.0%

0.0%

3.3

ANGIOEDEMA ABDOMEN

0.2%

0.1%

0.3%

0.0%

0.0%

0.1%

0.8%

0.0%

4.5%

0.0%

0.0%

3.3

DIFFUSE ITCHING

1.1%

0.8%

1.4%

0.5%

0.3%

0.8%

0.0%

0.0%

3.0%

0.9%

0.0%

4.9

GASTROINTESTINAL PAIN

5.6%

5.0%

6.2%

1.5%

1.2%

1.9%

3.3%

0.9%

8.2%

2.7%

0.6%

7.7

DIARRHOEA

4.6%

4.1%

5.2%

1.9%

1.6%

2.4%

3.3%

0.9%

8.2%

3.6%

1.0%

9.0

FAINT

1.4%

1.2%

1.8%

0.2%

0.1%

0.4%

0.8%

0.0%

4.5%

0.9%

0.0%

4.9

ORAL APHTOUS

1.3%

1.0%

1.6%

0.3%

0.2%

0.5%

0.0%

0.0%

3.0%

0.9%

0.0%

4.9

INSOMNIA

6.7%

6.1%

7.4%

1.6%

1.2%

2.0%

0.8%

0.0%

4.5%

7.2%

3.2%

13.7

ANOSMIA

0.3%

0.2%

0.4%

0.1%

0.0%

0.2%

0.0%

0.0%

3.0%

0.0%

0.0%

3.3

AGEUSIA

0.9%

0.7%

1.2%

0.1%

0.0%

0.2%

0.0%

0.0%

3.0%

0.0%

0.0%

3.3

PAIN

74.4%

73.3%

75.5%

51.9%

50.5%

53.3%

78.5%

70.1%

85.5%

85.6%

77.6%

91.5

ITCHING

5.5%

4.9%

6.1%

3.6%

3.1%

4.2%

6.6%

2.9%

12.6%

7.2%

3.2%

13.7

SWELLING

9.8%

9.0%

10.6%

5.6%

5.0%

6.3%

10.7%

5.8%

17.7%

11.7%

6.4%

19.2

REDNESS

6.4%

5.8%

7.1%

4.4%

3.8%

5.0%

6.6%

2.9%

12.6%

7.2%

3.2%

13.7
